# Improving Crop Lodging Resistance by Adjusting Plant Height and Stem Strength

**Yanan Niu [1], Tianxiao Chen [1], Chenchen Zhao [1] and Meixue Zhou [1,2,\***

[1] Tasmanian Institute of Agriculture, University of Tasmania, Private Bag 1375, Prospect, TAS 7250, Australia; yanan.niu@utas.edu.au (Y.N.); tianxiao.chen@utas.edu.au (T.C.); chenchen.zhao@utas.edu.au (C.Z.)
[2] College of Agronomy, Shanxi Agricultural University, Jinzhong 030801, China
[\*] Correspondence: meixue.zhou@utas.edu.au

**Abstract:** Crop height not only determines plant resistance to lodging and crowding, but also affects crop architecture, apical dominance, biomass, and mechanical harvesting. Plant height is determined by the internode elongation, regulated by genes involved in gibberellin (GA) and brassinosteroid (BR) biosynthesis or related signaling networks. Plants' genetic inability to synthesize or respond to GAs and BRs induce dwarfness. However, the signaling mechanisms of GAs and BRs for controlling plant height individually or collectively are still unclear. Since stem mechanically supports plant during the whole life span, components that affect stem physical strength are also important to crop lodging resistance. One of the major components is lignin, which forms stem structure, thus contributing to crop lodging resistance. In this review, we looked into the reported genes involved in lignin, GAs, and BRs biosynthesis and summarized the signaling networks centered by these genes. Then, we filled the knowledge gap by modifying plant height through interrupting normal GA and BR metabolism utilizing core gene inhibitors. Therefore, we highly endorsed the current approaches of using plant growth regulators (PRGs) to maintain an ideal plant height under lodging stress, and proposed possibilities of modifying crop culm strength against lodging as well.

**Keywords:** lodging resistance; plant height; GAs; BRs; lignin; PRGs





## 1. Introduction

Lodging refers to a permanent displacement of crops, stems or roots from their vertical orientation under unfavorable weather or soil conditions [1]. Therefore, lodging in cereals can be divided into stem lodging and root lodging [2,3]. Plant height is associated with stem lodging [4–6]. The final height of a plant is determined by internode elongation, which is regulated by genes involved in gibberellin (GA) and brassinosteroid (BR) biosynthetic or signaling pathways. As the two major plant hormones are involved in modulating diverse processes throughout plant growth and development, inadequate Gas and BRs biosyntheses lead to a dwarf or semi-dwarf stature, resulting in increased lodging tolerance [7–12]. In addition, interactions between GAs and BRs also regulate plant growth and development, but were not highlighted [13,14].

Plant growth regulators (PGRs) mainly function by regulating plant hormone biosynthesis, including GAs and BRs, or related signal transductions in cereal crops for maintaining targeted agronomic traits, such as crop height to secure cereal yield. Even though an increasing number of PGRs are being applied for multiple agricultural purposes, the most common purpose is still to reduce crop height [15].

The biochemical characteristics of stems, such as cellulose, hemicellulose, lignin, silica, and soluble sugar contents, which are classified as structural carbohydrates (SC), significantly contribute to stem physical strength against lodging stress. Low lignin or cellulose contents in the stem result in brittleness of culm of plants, such as *Arabidopsis* [16], rice [17,18] and buckwheat [19] and lodging-resistant varieties show more lignin accumulation than lodging-susceptible ones [20–22].

GAs, BRs and lignin pathways have been subjected to intensive studies in *Arabidopsis*, but the genes and orthologs or homologs involved in the biosynthetic or signaling processes have not been fully discovered in major staple crops, such as rice, wheat, barley, and maize.

## 2. Biosynthetic Pathways Involved in Plant Height Regulation

*2.1. GA Biosynthesis or Signaling Pathways*

Gibberellins (GAs) play an important role in modulating diverse processes throughout plant growth and development, mainly stem elongation. Internode elongation is an important agronomic trait that determines final culm length, panicle exertion, and crop biomass [23]. Ample evidences from rice, barley, and *Arabidopsis* mutants indicate a common mechanism that internode elongation is regulated by genes involved in gibberellin (GA) and brassinosteroid (BR) biosynthetic or signaling pathways [24]. This can be evidenced either by mutants with decreased bioactive GA concentrations which leads them to be dwarf or semi-dwarf in stature, or elevated bioactive GA concentrations which leads to increased crop height [7]. Semi-dwarf rice mutants with gibberellic acid (GA)-deficiency or GA-insensitivity are more tolerant to lodging stress under extreme environmental conditions, indicating that lodging tolerance can be increased by decreasing plant height through phytohormone GA accumulation [8,9]. Due to the property of being short-statured, plants lacking GAs displayed higher bending-type lodging resistance but lower breaking-type lodging resistance [25].

Three main stages are involved in the isoprenoid pathway leading to GA biosynthesis (Figure 1) [26,27]. The first stage starts with mevalonic acid and other metabolites (Figure 1). Isopentenyl diphosphate (IPP), farenesyl pyrophosphate (FPP) and geranylgeranyl pyrophosphate (GGPP) are the key intermediates, which are also precursors for cytokinin, abscisic acid (ABA), sterol, terpenoid and carotenoid biosynthesis [26,28]. The second stage is a series of oxidation reactions that generate GA12-aldehyde, which is a specific intermediate for GA formation (Figure 1). The final step is the synthesis of active GAs, which is the catalysation of 2-oxoglutarate-dependent dioxygenases, including 20-oxidase (20ox) and 3-oxidase (3ox) (Figure 1). The enzymes involved in this process are listed in Figure 1.

The first characterization of a GA-biosynthetic mutation was reported in maize, in which *dwarf-5* mutant was defective in ent-kaurene synthase (KS) activity, producing ent-isokaurene rather than ent-kaurene [29]. Two cytochrome P450 genes, *CYP714B1* and *CYP714B2*, encode GA 13-oxidases, which plays a role in fine-tuning plant growth by decreasing GA bioactivity, and overexpression of these two genes in rice induced semi-dwarfism [30]. Correspondingly, most mutants or knockdown lines of GA biosynthesis genes, including *CPS* (*ent-copalyl diphosphate synthase*), *KS*, *KAO* (*ent-kaurenoic acid oxidase*), *KO* (*ent-kaurene oxidase*), *GA20oxs*, and *GA3oxs*, also exhibit dwarfism phenotypes, which results in improved lodging resistance, a valuable trait for rice breeding [31,32]. However, some mutants become extremely dwarfed, thus cannot be used in breeding programs.

Genes encode *20-oxidase* in GA biosynthetic pathways were identified in various cereal crops, affecting the later steps in GA biosynthetic pathway, thereby reducing plant height. For example, rice *sd-1* mutants and *sdw1* (*HvGA20ox2*) allele in barley have loss-of-function mutations in the GA synthesis gene, *GA 20 oxidase2* (*semi dwarf-1*; *SD-1*) [33,34], while wheat *Reduced height-1* (*Rht-1*) mutants have gain-of-function mutations in a gene encoding a suppressor of a GA signal known as the DELLA protein [35]. The wheat Green Revolution genes, *Rht-B1b* (*Rht1*) and *Rht-D1b*(*Rht2*), are orthologues of the *Arabidopsis gibberellic acid-insensitive* (*gai*) [36], the maize *dwarf-8* (*d8*) [37], the rice *OsGAI* also known as *SLR1* [38,39], and the barley slender1 (*sln1*) genes [40]. A 17-amino acid deletion affecting the DELLA region resulted in GA-insensitive dwarf rice phenotypes [38], which are similar to that in the *gai* mutant of *Arabidopsis* [41].

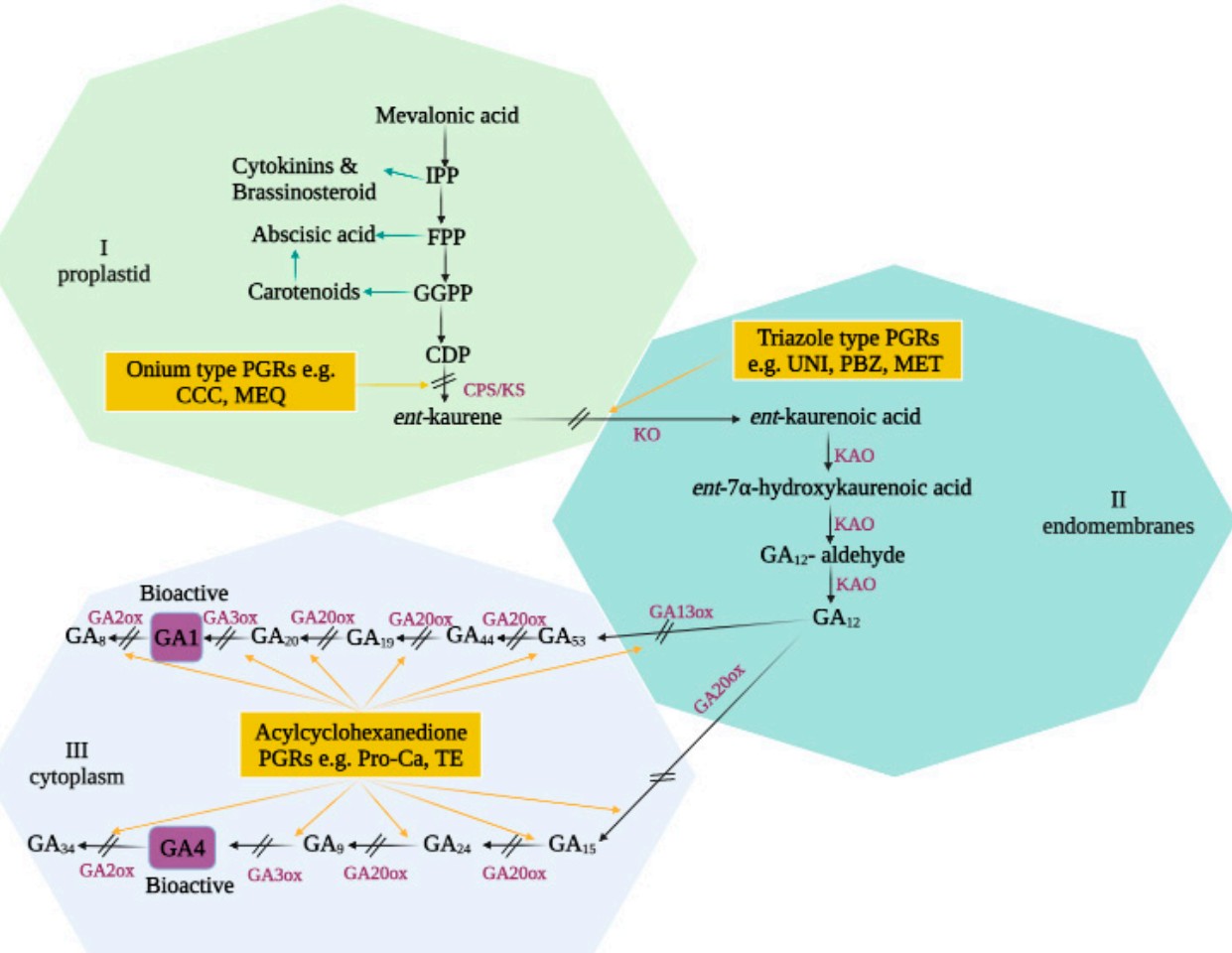

**Figure 1.** GA biosynthetic pathways in cereals. The respective enzymes involved in this process are in purple. Arrows with oblique lines show the inhibition of PGRs. Anti-GA PGRs are in the yellow rectangle. Blue arrows show the crosstalk with other plant hormones. IPP: Isopentenyl diphosphate; FPP: Farenesyl pyrophosphate; GGPP: Geranylgeranyl pyrophosphate; CDP: *ent*-Copalyl diphosphate; CPS: *ent*-copalyl diphosphate synthase; KS: *ent*-kaurene synthase; KO: *ent*-kaurene oxidase; KAO: *ent*-kaurenoic acid oxidase.

As GA positively regulates the diameter of culm internodes and lignin content, it improves lodging resistance in two perspectives in rice; structurally, by increasing culm diameter, and qualitatively, by increasing lignin content [25]. Compared to BR, GA has a relatively minor role in promoting coleoptile and root elongation in rice [42]. Exogenous GA greatly promoted leaf sheath elongation in rice, resulting in sheaths that were up to 4-fold longer than those not treated with GA [43].

Overproducing *GA 2 oxidase* (*GA-inactivating enzyme*) exhibited increased tillering by negatively regulating expression of *Os TEOSINTE BRANCHED1* (*OsTB1*), a positive regulator for strigolactone signaling [44]. The ABA compound can be produced directly from FPP or indirectly from the conversion of GGPP to carotenoids, and cytokinin compound can be produced directly from IPP (Figure 1) [26]. Inhibited GA-inducible responses by ABA, such as the expression of α-amylase by regulating *WRKY* transcription factors, are also reported in rice [45]. Commercial PGRs, such as chlormequat (CCC) and mepiquat (MEQ), which are also defined as onium-type PGRs, have been found to decrease height and increase stem diameter in cereals by blocking the activity of specific enzymes involved in GA biosynthesis (Figure 1) [27,28]. PBZ (paclobutrazol) and UNI (uniconazole), which are classified as triazole compounds, are highly efficient in binding and inactivating enzymes

involved in the conversion of ent-kaurene to ent-kaurenoic acid (Figure 1) [27]. Acylcyclohexanedione PGRs such as Pro-Ca (prohexadione-calcium) and TE (trinexapac-ethyl) have a similar structure to 2-oxoglutaric acid, therefore inhibit the formation of activated GA (Figure 1) [27,46,47].

### 2.2. BR Biosynthetic or Signaling Pathways

Brassinosteroids (BRs) are a class of steroidal phytohormones that play an essential role in regulating diverse processes during the whole life cycle of plants and plants' adaptation to abiotic stresses [48,49]. Based on the total number of carbons, BRs are divided into C27, C28, and C29-type. Three pathways of BR biosynthesis leading to the production of $C_{27}$-, $C_{28}$-, or $C_{29}$-type of BRs are currently known in *Arabidopsis thaliana* [50] (Figure 2). Early steps of their synthesis are common for each type and may occur via a mevalonate (MVA) or non-MVA pathway, while later steps differentiate BR biosynthesis pathways [51]. The direct pathway is $C_{27}$-BRs, from IPP to cholestanol, and then to 28-norBL (Figure 2). The biosynthesis of $C_{29}$-BRs is initiated from β-sitosterol and leads to 28-homoBL (Figure 2). $C_{28}$-BRs biosynthesis pathway starts from episterol to sampesterol, and then goes to $C_{28}$-BRs (Figure 2). The last step in the transformation of castasterone (CS) to BRs is not clear in the dicotyledons (Figure 2). So far, most of the reactions, enzymes, and genes have only been discovered and characterized by the $C_{28}$-BR biosynthesis pathway in *Arabidopsis thaliana*, while fewer genes were isolated in cereals [52].

Several components of the BR signaling pathway in rice, such as *OsBRI1* (*Brassinosteroid-Insensitive1*) [53], *OsBAK1* (*BRI1-Associated receptor Kinase1*) [54] form a core of the transmembrane BR receptor complex, *OsGSK1* and *OsGSK2* (*Glycogen Synthase Kinases*) [55]. The complex is a major negative regulator of BR signaling, and *OsBZR1* (*Brassinazole-Resistant1*) transcription factor plays a pivotal function in BR-dependent regulation of gene expression [56]. BR-deficient mutants usually display decreased leaf length, erect leaves, reduced plant height, and shortened roots, and exogenous BR application could have inhibitory effects on rice growth and development [42]. Mutations in the *OsBRI1* gene (*d61*) in rice were loss-of-function mutations. The mutants showed semi-dwarfism, erect stature and BR insensitivity [10,11]. Overexpressing the wheat *TaBRI1* gene in *Arabidopsis* led to faster germination, early flowering, and higher seed yield [57]. In barley, a series of alleles of the homologous gene, *HvBRI1,* has been identified. One of the alleles, *uzu1.a*, is a well-known semi-dwarfing allele in Northeast Asian short-culm cultivars and landraces [58]. A loss-of-function mutation of the *OsBAK1* gene results in erect leaves and the BR insensitivity, but without any significant effect on plant height, reproduction and grain yield [11]. In wheat, the homologs of *OsBAK1* are the *SERK* family proteins. Functional analysis indicated that *TaSERK* genes in *Arabidopsis* led to increased height and seed yield [59].

On the other hand, mutants or transgenic rice plants with enhanced BR levels or BR signaling have been observed to display reduced plant height. For example, CYP724B1/D11 is involved in the brassinosteroid biosynthesis pathway [60], BRASSINOSTEROID UP-REGULATED1(BU1) overexpressor (Tanaka et al., 2009) [61], *BAK1* and [62], DWARF AND LOW-TILLERING (DLT) overexpressor, which work in the signal transduction pathway [63]. Evidence has also been found in *Arabidopsis* with low concentrations of BR promoting the growth of both the root and hypocotyl, whereas high concentrations of BR inhibit root growth but still promote hypocotyl growth (Müssig et al., 2003). Similarly, in rice, BR significantly promotes coleoptile growth, but a relatively higher concentration of BR inhibits both root and seedling growth (Tong et al., 2009). *HvD1*(*Brh1*) and *HvDEP1* are two barley genes encoding the α-subunit and γ-subunit of the heterotrimeric G protein, respectively. *HvD1*(*Brh1*) caused a semi-dwarf phenotype, but did not show any major negative impact on malting quality, which is a very important trait in barley breeding [12]. Loss of function in a cytochrome P450 (*CYP90B2*) involved in BR biosynthesis in rice was detected in the *Osdwarf4-1* mutant which exhibits erect leaves and slight dwarfism without compromising grain yield [64].

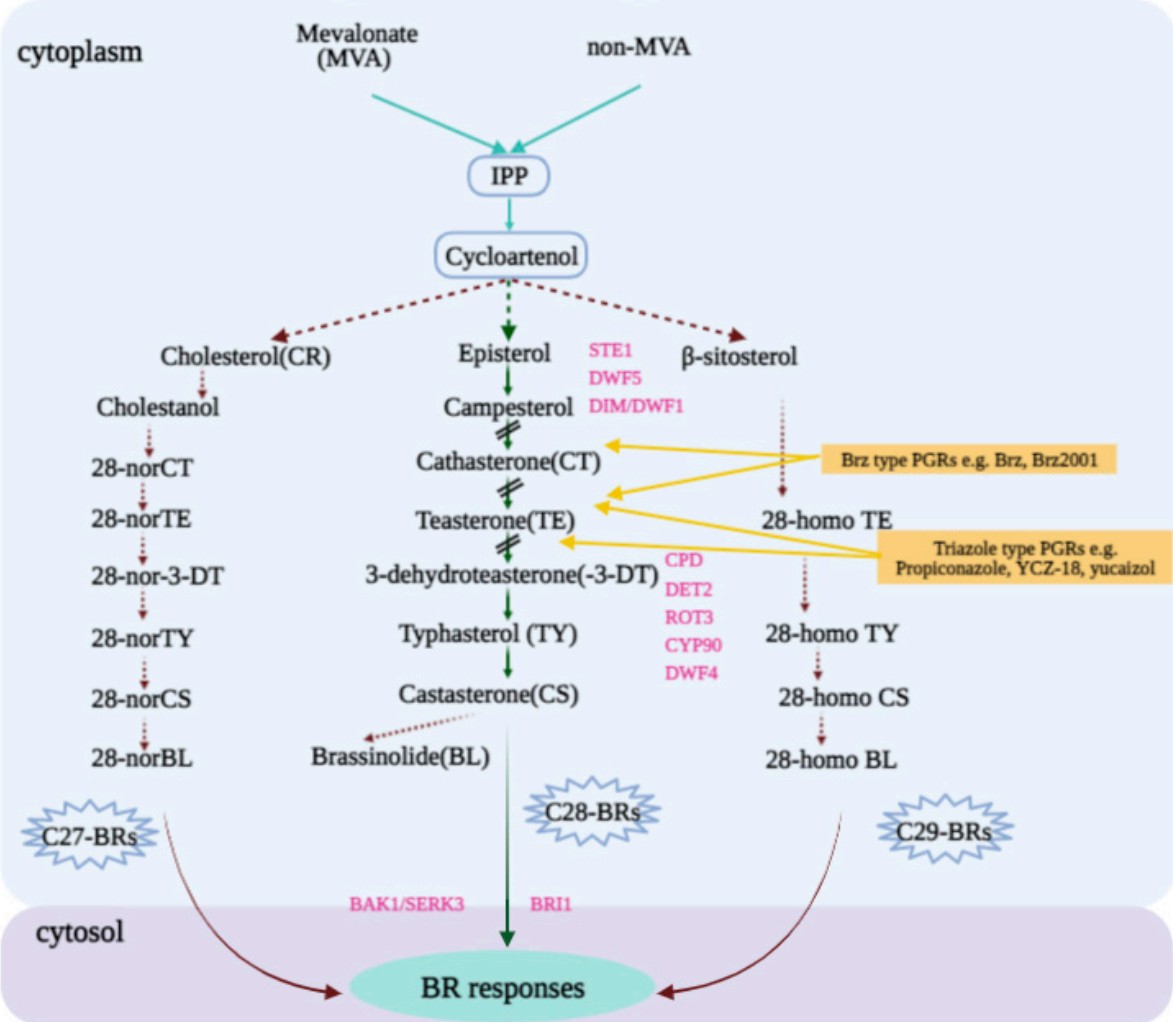

**Figure 2.** Brassinosteroid synthesis pathway in cereals. Blue lines are the common steps both for cereals and *Arabidopsis*, brown lines show the steps in *Arabidopsis*, and green lines show the steps found in cereals. The respective enzymes involved in this process are in purple. The final product of the pathway is Castasterone (CS). Green arrows with oblique lines show the inhibition of PGRs. Anti-GA PGRs that inhibit steps are shown in the yellow rectangle. IPP: Isopentenyl diphosphate; *STE1*: $\Delta^7$-sterol-C5-desaturase; *DWF5*: $\Delta^{5,7}$-sterol-$\Delta^7$-reductase; *DIM/DWF1*: $\Delta^5$-sterol-$\Delta^{24}$-reductase; *DWF4*: C-22 hydroxylase; *DET2*: 5α reductase; *CPD*: C-23α-hydroxylase/C-3 dehydrogenase; *ROT3* and *CYP90D1*: C-23 hydroxylases; *BR6ox1*: brassinosteroid-6-oxidase 1; *BR6ox2*: brassinosteroid-6-oxidase 2.

BRs also interact with other phytohormones to regulate plant growth and development [13]. An early study found that BRs and GAs act antagonistically to regulate the expression of a GA-responsive gene, *GASA1* (for GA-stimulated transcript in *Arabidopsis*), as well as a GA biosynthetic gene, *GA5* (GA20ox) [14]. In another study, BR was found to induce several GA biosynthetic genes, including *GA20ox-1*, *GA20ox-2*, and *GA20ox-5*, in *Arabidopsis* [65]. GA-deficient or GA-insensitive mutants are sensitive to BR, whereas a mutant lacking DELLA proteins has greatly enhanced BR sensitivity [66].

The rice *D1* gene encodes heterotrimeric G-protein alpha subunit (*RGA1*) which functions in several signaling pathways [67]. Mutations in the *D1* gene lead to characteristic BR-specific phenotype, including reduced height and erect leave. It is also known that mutations in the *OsD1/RGA1* gene affect GA signal transduction and disease resistance [68], thus *OsD1/RGA1* is involved in crosstalk between the BR and GA signaling pathways [11]. Rice BR-GA hormonal crosstalk is evidenced by the fact that *OsBZR1* directly binds to promoters of the *GA20ox-2*, with *GA3ox-2* (GA biosynthetic genes) greatly inducing their expression and *GA2ox-3* (GA inactivation gene) repressing its transcription [49]. *SPINDLY*

participates in both BR and GA responses, thus it regulates the elongation of lower internodes of rice [69]. However, in rice root, BR application appears to repress the levels of active GA by inhibiting the expression of *GA20ox-3*, a GA biosynthetic gene, and by simultaneously promoting the expression of *GA2ox-3*, a GA inactivation gene [70]. Generally, BR promotes GA biosynthesis and inhibits GA inactivation, while GA extensively inhibits BR biosynthesis and BR response as a feedback mechanism in rice [42,70]. Overexpression of *OsIAA1* in rice, which is a member of the Aux/IAA family proteins, results in reduced auxin sensitivity but increased sensitivity to BR [71]. In a previous study, Liu et al. [72] summarized the dwarf genes involved in interactions between the other relevant dwarfing phytohormones, which facilitates our understanding of crosstalk between different hormone pathways.

So far, 17 inhibitors for BR pathway—KM-01, brassinozole (Brz), Brz2001, Brz220, propiconazole, YCZ-18, yucaizol, fenarimol, spironolactone, triadimefon, imazalil, 4-MA, VG106, DSMEM21, finastride, AFA76, and brassinopride—have been discovered in *Arabidopsis thaliana* [73]. Inhibitors used in the $C_{28}$-BRs biosynthesis pathway are Brz, Brz2001, Brz220, propiconazole, YCZ-18, yucaizol, and fenarimol [52] (Figure 2). The chemical structure of Brz is similar to triazole-type PGRs such as UNI and PBZ, which block the conversion of campestanol to TE, and Brz2001 has the same function as Brz [74]. The triazole compound, propiconazole, blocks the same reactions with Brz, and YCZ-18 and yucaizol bind to the CYP90D1 enzyme and inhibit the BR-induced cell elongation [75]. Other PGRs such as fenarimol inhibit the conversion of CT to TE [76].

These results provide strategies for genetic improvement and field management of crop production by modulating BR biosynthesis and signal transduction, and the crosstalk of other hormones. With a better understanding of the hormonal regulation of culm elongation, a similar strategy would also be possible for other components in BR biosynthesis and signaling pathways.

## 2.3. Lignin Biosynthesis Mechanism

The biochemical pathways of monolignol biosynthesis are highly conserved throughout vascular plants, and most current research has focused on monolignol biosynthesis. However, although a majority of enzymes in the monolignol biosynthesis pathway have been identified and characterized, additional pathway components cannot be ruled out [77]. It is well known that peroxidases and laccases are involved in dimerization and cross-linking of monolignols, but more detailed mechanisms have yet to be unveiled [78]. Moreover, multifunctional enzymes involved in lignin biosynthesis pathways also correspond to diverse gene families, such as *COMT*, which can be used as an elicitor-induced plant defense response, and *F5H* which acts as a cytochrome-P450-dependent monooxygenase [77]. Hence, it is a big challenge to explore the molecular mechanism behind the lignin biosynthesis pathways.

Phenylalanine ammonia lyase (*PAL*), cinnamate 4-hydroxylase (*C4H*), 4-coumaroyl-CoA ligase (*4CL*), chalcone synthase (*CHS*), caffeoyl-CoA 3-*O*-methyltransferase (*CCoAOMT*), and hydroxycinnamoyl transferase (*HCT*) are key genes in the biotechnological alteration of lignin biosynthesis to improve wood properties [79,80]. Similarly, in wheat, CoA ligase1 (*4CL1*), cinnamoyl-CoA reductase2 (*CCR2*), ϱ-coumarate 3-hydroxylase1 (*C3H1*), ferulate 5-hydroxylase2 (*F5H2*), and caffeic acid *O*-methyltransferase2 (*COMT2*) were highly expressed in wheat tissues, indicating the significance of these genes in the intervening lignin accumulation in wheat culm [81] (Figure 3).

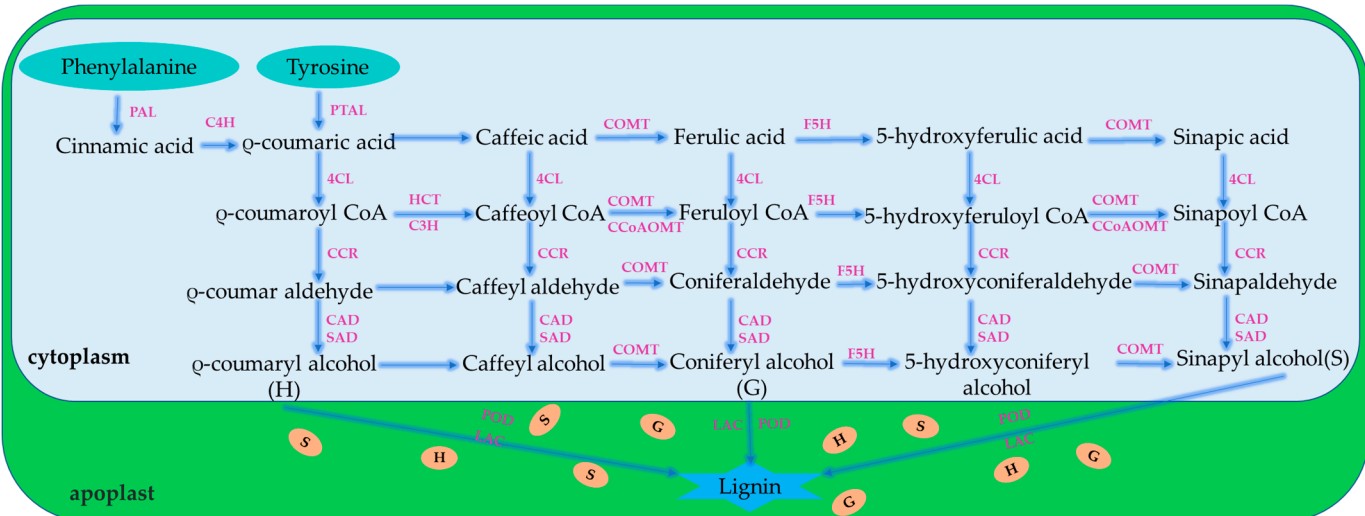

**Figure 3.** Lignin biosynthetic pathway in cereals. Enzymes involved in this pathway include *PAL, PTAL, C4H, 4CL, HCT, C3H, COMT, CCoAOMT, CCR, CAD, F5H, POD* and *LAC*. The respective enzymes involved in this process are in purple. PAL: phenylalanine ammonia lyase; PTAL/TAL: phenylalanine /tyrosine ammonia-lyase; C4H: cinnamate4-hydroxylase; 4CL: 4-coumarate: coenzyme A ligase; HCT: hydroxycinnamoyl CoA shikimate/quinate hydroxycinnamoyl transferase; C3H: P-coumarate 3-hydroxylase; COMT: caffeic acid *O*-methyltransferase; CCoAOMT: caffeoyl-CoA *O*-methyltransferase; CCR: cinnamoyl-CoA reductase; CAD: cinnamyl alcohol dehydrogenase; F5H: ferulate5-hydroxylase; POD: peroxidase; LAC: Laccase. S, G, and H stand for sinapyl alcohol, coniferyl alcohol, and *p*-coumaryl alcohol lignin units, respectively.

*CCR* is a key gene in the lignin monomer biosynthesis pathway. Under abiotic stress condition, *CCR1/2* were significantly up-regulated in the root elongation region of maize [82]. Studies on lodging-resistant (H4546) and lodging-sensitive (C6001) wheat cultivars suggested that *Ta-CCR1* [83], *TaCAD1* [84], *TaCCoAOMT1* [85], and *TaCM* [86] and their related enzymes are involved in lignin biosynthesis and are critical for lodging resistance. During the biosynthesis of lignin monomers (or monolignols), the formation of sinapyl alcohol requires the 5-O-methylation of 5-hydroxy to conifer aldehyde catalyzed by caffeate *O*-methyltransferase (*COMT*, EC 2.1.1.68) [87,88]. Similar results have been found in rice (*OsCOMT1*) [89,90], maize (*ZmCOMT*) [91], barley (*HvOMT1*) [91], and sorghum (*SbCOMT*) [92]. PALs can catalyze the lignin precursor phenylalanine and transform it into cinnamic acid in the lignin biosynthesis pathway [93]. In rice, overexpression of an *F5H* gene *OsCAld5H1* increased the content of S units, while down-regulation of this gene enhanced the production of G lignin [94]. AMP-binding domain-containing *4CLs* are critical enzymes in phenylpropanoid metabolism pathway with the loss of *4CL1* leading to reduced lignin content in *Arabidopsis* [95]. Suppression of *Os4CL3* expression results in significant lignin reduction, impaired plant growth, decreased panicle fertility, and reduced height of rice [96]. *OsAAE3* is a homolog of *Arabidopsis AAE3* in rice, which encodes a 4-coumarate-Co-A ligase (*4CL*) such as protein. Over-expression of *OsAAE3* resulted in a significant decrease in expressions of lignin biosynthesis genes, leading to reduced lignin content [97]. Cinnamyl alcohol dehydrogenase (*CAD*) catalyzes the last step of monolignol biosynthesis. *OsCAD2* is largely responsible for monolignol biosynthesis in rice stem, while mutant plants exhibit drastically reduced *CAD* activity and undetectable sinapyl alcohol dehydrogenase activity [98,99]. In maize, a *CCoAOMT* gene *ZmCCoAOMT2,* which is associated with resistance to multiple pathogens, is involved in the biosynthesis of lignin and other phenylpropanoid metabolites and regulation of programmed cell death [100]. However, in wheat, *TaCCoAOMT1* is critical for stem development but there is no evidence to show that it is directly associated with lodging-resistance [85]. Two *HCT* genes, *HCT1806*, and *HCT4918*, were identified in maize and regulate plant disease resistance by binding to NLR Rpl protein, which enhanced the expression of lignin biosynthesis pathway genes and lignin accumulation [101]. All of the above suggests that the lignin biosynthesis pathway

and its related enzyme biosynthesis play a vital role in assisting plants in stronger stem strength or soil anchorage, and these traits ultimately make them tolerant to lodging stress environment.

General processes involved in the lignin biosynthesis pathway in plants have been illustrated in previous studies [102,103]. However, the functions of these genes involved in the lignin biosynthesis pathway remain unexplored in many cereal crops, such as acyl-coenzyme synthetases (LACSs) [104]. A 20–40% reduction in lignin content has been observed in knockout mutants of two laccase genes, *LAC4* and *LAC17* [105], and a complete loss of lignin deposition in roots has been detected in a triple mutant of *LAC4*, *LAC11*, and *LAC17* in *Arabidopsis* [106]. Here, we present a general lignin biosynthesis pathway in cereals (Figure 3). It is extremely important to have a full understanding of lignin biosynthesis pathway, as it plays a vital role in reducing financial losses caused by internal and external factors and guiding researchers to develop better strategies for crop yield improvement.

## 3. Future Perspectives

Crop height is an important factor for lodging resistance, fundamentally affect crop yield [72]. The "green revolution", which benefited from the breeding of semi-dwarf crops, along with proper applications of fertilizer and pesticide, has greatly increased crop production [107,108]. While there is no doubt about the contribution of semi-dwarfing, extreme dwarfing leads to small grains, semi-sterility, malformed panicles, thus decreased yield and biomass production [72,109,110]. Therefore, it is necessary to exploit dwarf germplasm, and identify novel semi-dwarf genes without adverse effects on agronomic traits.

The BR-responsive module, *OsmiR159d-OsGAMYBL2*, acts as a common component functioning in both BR and GA pathways which connect BR signaling and GA biosynthesis, and thus coordinate the regulation of BR and GA in plant growth and development [111]. Castorina and Consonni (2020) presented a model to illustrate that BR promotes GA biosynthesis and inhibits GA inactivation, which leads to increased GA levels and cell elongation. Dwarf stature caused by mutants involved in the BR pathway is attributed to reduced internode length but not decreased internode number [112]. By binding to the promoters of GA biosynthetic genes, BR could modulate the activities of growth-related genes through the interaction with GAs. Mechanisms of BR-GA interaction in controlling plant height has been well studied both in *Arabidopsis thaliana* [113] and in rice [42], hence, it is worth exploring the function of BR-GA interaction in dwarfing for future applications in other cereals.

Even though dwarfing was mainly attributed to the plants' genetic inability to synthesize or respond to GAs and BRs, other hormones, including strigolactones (SLs) [114], indole-3-acetic acid (IAA) [115], and abscisic acid (ABA) [116] are also associated with crop height. Plant height is controlled by genes that formed a complex regulatory network, mainly involving the biosynthesis or signal transduction of phytohormones. Excessive focus on the application of dwarf genetic materials such as *sd1* and *Rht1* poses a high risk of losing genetic diversity. Therefore, other genes involved in the GA and BR pathways need to be exploited. It is necessary to not only understand the effect of newly identified genes in crop height, but also investigate its effect on other traits related to lodging. In addition, whether these identified genes involve GA and BR synthesis which can be utilized to optimize crop lodging should also be investigated. In addition, environmental factors such as light, temperature, water, and nutrition also need to be ascertained in order for appropriate crop height-related genes to be used in different ecological areas.

Lignins are important components of the secondary cell wall, which decide cell wall stiffness and mechanical support to the plant body, enabling plants to grow upwards [117,118]. To overcome the low grain production caused by dwarf plant architecture, most studies have focused on increasing plant density for securing crop production. However, severer lodging was observed under higher plant densities and this is mostly due to the lignin synthesis [119–121]. Lignin accumulation and its composition (i.e., H-, G-

and S-type monomers) are important factors influencing the breaking strength of crop culm [121]. How plant density regulates lignin biosynthesis in the basal culm, and its relationship with lodging resistance in cereals are of great importance. A reasonable plant density is needed to decrease the risk of lodging occurring, not only by altering the basal stem morphological traits but also by modifying lignin metabolism. As lignin deposition in the plant cell walls can also be affected by environmental conditions such as biotic (bacteria, fungi and virus) and abiotic stresses (mineral deficiency, drought, ultraviolet-B (UV-B) radiation and low temperatures), and mechanical injuries, a greater understanding of lignin biosynthesis under different agricultural environments should be researched [121,122].

Lignin is the predominant cell wall polymer that significantly enhances cell wall thickness, thereby increases stem breaking force for high lodging resistance in rice [123]. Although low cellulose content along with a significant decreased cell wall thickness were observed in the dwarf barley plants, over-expression of barley secondary cell wall cellulose synthase (*HvCesA*) shows no increases in overall crystalline cellulose content or stem strength [124]. Compared with lignin biosynthesis, the individually up-regulated cellulose level in plants is likely to require more sophisticated strategies in the future. Therefore, increasing lignin level rather than cellulose or hemicelluloses becomes critical for high cell wall strength. Meanwhile, genetic modification of crop cell walls has a great potential for improving crop lodging resistance. However, we have to consider whether lignin affects crop height during the process of regulating crop resistance against lodging.

**Author Contributions:** Conceptualization, Y.N., T.C. and M.Z.; Visualization, Y.N. and T.C.; Writing—Original Draft Preparation, Y.N.; Writing—Review and Editing, M.Z. and C.Z.; Supervision, M.Z.; Project Administration, M.Z.; Funding Acquisition, M.Z. All authors have read and agreed to the published version of the manuscript.

**Funding:** This work is funded by the Grains Research and Development Corporation (GRDC) of Australia (PROC-9175606).

**Institutional Review Board Statement:** Not applicable.

**Informed Consent Statement:** Not applicable.

**Data Availability Statement:** Not applicable.

**Conflicts of Interest:** The authors declare no conflict of interest.

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
