# Peer review of "Improving Crop Lodging Resistance by Adjusting Plant Height and Stem Strength"

_agronomy, doi:10.3390/agronomy11122421_

Round 1

Reviewer 1 Report

This manuscript outlines the biosynthesis and signaling of gibberellins, brassinosteroids, and lignin in the context of reducing crop lodging. The authors also summarize the biosynthesis inhibitors for gibberellins and brassinosteroids. This manuscript may be useful in understanding the control of plant height by dwarf mutants or plant growth regulators. On the other hand, the chapters on gibberellins, brassinosteroids, and lignin are independent and sporadic, and do not seem to have been comprehensively discussed. Below I have a number of suggestions that the authors may consider in the further process of improving the manuscript. The suggestions are listed in order of appearance and not in order of priority.

P1, lines 34-35: ‘inadequate GAs and BRs expressions’

Does this refer to gene expression or biosynthesis? The word "expression" is ambiguous.

P2, lines 80-81: From this text alone, readers who are not familiar with GAs cannot understand why plants overexpressing GA13ox, a gene involved in biosynthesis, become semi-dwarf. The authors need to add an explanation.

P2, lines 82-85: As the authors state in future perspectives, not all mutants are useful for agriculture, and some mutants become extremely dwarf. However, this sentence gives the impression as if all mutants are valuable for agriculture. Also, some mutants have severe or mild phenotypes depending of the allele. The authors should add explanations for these points.

P2, line 93: ‘the rice gai or slender1’. This makes it sound like there are two genes, but they are the same gene. If you are referring to SLRL1, please change the wording to something more appropriate. In addition, there are two phenotypes of slr1 mutant depending on the mutation position: a slender mutant and a dwarf mutant. I think this needs to be explained to show the dwarfism of DELLA proteins.

P3, line 111: ‘from IPP (Figure 3)’. Is this Figure 1?

P5, line194: ‘involved in’ was repeated.

In Future perspective.

Arabidopsis and pea brasinosteroid mutants are not elongated by gibberellins, and gibberellin-deficient mutants are less responsive to brassinosteroids. Thus, this indicates that both brassinosteroids and gibberellins are necessary for elongation, but their actions are independent of each other and not complementary. The authors can include these findings and describe future prospects for dwarfing using brassinosteroids and gibberellins.

Author Response

Thank you for your constructive suggestions. We have revised the MS according to your suggestions. Please see details below:

This manuscript outlines the biosynthesis and signaling of gibberellins, brassinosteroids, and lignin in the context of reducing crop lodging. The authors also summarize the biosynthesis inhibitors for gibberellins and brassinosteroids. This manuscript may be useful in understanding the control of plant height by dwarf mutants or plant growth regulators. On the other hand, the chapters on gibberellins, brassinosteroids, and lignin are independent and sporadic, and do not seem to have been comprehensively discussed. Below I have a number of suggestions that the authors may consider in the further process of improving the manuscript. The suggestions are listed in order of appearance and not in order of priority.

Thanks. 

P1, lines 34-35: ‘inadequate GAs and BRs expressions’

Does this refer to gene expression or biosynthesis? The word "expression" is ambiguous.

 Response: Thanks for pointing out this.  We have revised it to “inadequate GAs and BRs biosyntheses”.

P2, lines 80-81: From this text alone, readers who are not familiar with GAs cannot understand why plants overexpressing GA13ox, a gene involved in biosynthesis, become semi-dwarf. The authors need to add an explanation.

 Response: Explanation has been added to this section.

P2, lines 82-85: As the authors state in future perspectives, not all mutants are useful for agriculture, and some mutants become extremely dwarf. However, this sentence gives the impression as if all mutants are valuable for agriculture. Also, some mutants have severe or mild phenotypes depending of the allele. The authors should add explanations for these points.

 Response: We have added an extra sentence to the end of this paragraph - lines 85-89 (original 82-85).

P2, line 93: ‘the rice gai or slender1’. This makes it sound like there are two genes, but they are the same gene. If you are referring to SLRL1, please change the wording to something more appropriate. In addition, there are two phenotypes of slr1 mutant depending on the mutation position: a slender mutant and a dwarf mutant. I think this needs to be explained to show the dwarfism of DELLA proteins.

 Response: Thanks for your suggestions, changes have been made to this section.

P3, line 111: ‘from IPP (Figure 3)’. Is this Figure 1?

 Response: Thanks, we have fixed it.

P5, line194: ‘involved in’ was repeated.

 Response: Corrected.

In Future perspective.

Arabidopsis and pea brasinosteroid mutants are not elongated by gibberellins, and gibberellin-deficient mutants are less responsive to brassinosteroids. Thus, this indicates that both brassinosteroids and gibberellins are necessary for elongation, but their actions are independent of each other and not complementary. The authors can include these findings and describe future prospects for dwarfing using brassinosteroids and gibberellins.

Response: Following your suggestions, we have added on paragraph to this section.

Reviewer 2 Report

Overall, I found this to be a well-written and compelling review article linking phytohormones and cell wall composition to plant height and lodging resistance. I have a few points to improve the clarity of the manuscript.

  1. The authors have generally done a good job of indicating the plant species used for each result discussed, however there are a couple of instances where the plant species is not noted.  e.g. Lines 45-47; Lines 101-103; others. The authors should check that each result described includes the species.
  2. The title for section 2, "Pathways involved in crop lodging resistance" is somewhat misleading. Lodging resistance is the outcome of multiple intersecting factors. Pathways leading to shorter plants are not always lodging-resistant if the management strategies are incompatible. The authors should consider re-wording this title.
  3. The representation of the biosynthetic pathways is a great addition to the paper. However, I would suggest that there is a common theme across Figures 1-3. For example, in Fig 2, please add the sub-cellular compartment(s). Also suggested to keep the color of the sub-cellular compartments the same across all figures for cohesiveness.
  4. I would recommend the authors move the cross-talk paragraph into it's own section. At the moment it gets lost in the discussion of BR.
  5. In Lines 45-46 the authors highlight results that low lignin OR low cellulose result in stem brittleness, but in the article (and in particular the Future Perspectives section), the authors emphasize the importance of focusing on lignin. Can the authors expand upon this discussion of why low cellulose results in stem brittleness, but they don't believe it is an important target?   

Author Response

Thanks for your constructive suggestions. Please see below detailed revision notes with revised parts in red font:

Overall, I found this to be a well-written and compelling review article linking phytohormones and cell wall composition to plant height and lodging resistance. I have a few points to improve the clarity of the manuscript.

  1. The authors have generally done a good job of indicating the plant species used for each result discussed, however there are a couple of instances where the plant species is not noted. g. Lines 45-47; Lines 101-103; others. The authors should check that each result described includes the species.

Response: We very appreciate your feedback, we have now added the plant species where needed, e.g. Lines 46-47, Lines 105-108.

  1. The title for section 2, "Pathways involved in crop lodging resistance" is somewhat misleading. Lodging resistance is the outcome of multiple intersecting factors. Pathways leading to shorter plants are not always lodging-resistant if the management strategies are incompatible. The authors should consider re-wording this title.

Response: We have re-worded this title as “Biosynthetic pathways involved in plant height regulation”

  1. The representation of the biosynthetic pathways is a great addition to the paper. However, I would suggest that there is a common theme across Figures 1-3. For example, in Fig 2, please add the sub-cellular compartment(s). Also suggested to keep the color of the sub-cellular compartments the same across all figures for cohesiveness.

Response: Modifications have been made to Figures 1-3 according to the suggestions.

  1. I would recommend the authors move the cross-talk paragraph into it's own section. At the moment it gets lost in the discussion of BR.

Response: Good suggestion, however, we would like to keep this part in discussion to show potential links between different plant hormones. Another reason for not separating them is that it is too short to form a section.

  1. In Lines 45-46 the authors highlight results that low lignin OR low cellulose result in stem brittleness, but in the article (and in particular the Future Perspectives section), the authors emphasize the importance of focusing on lignin. Can the authors expand upon this discussion of why low cellulose results in stem brittleness, but they don't believe it is an important target?

Response: We have added more to the Future Perspectives section.